# Positional Encoding Helps Recurrent Neural Networks Handle a Large Vocabulary

**Takashi Morita**                                             *tmorita@alum.mit.com*
*Academy of Emerging Sciences | Center for Mathematical Science and AI*
*Chubu University*

**Reviewed on OpenReview:** *https://openreview.net/forum?id=PtnwXd13SF*

## Abstract

This study reports an unintuitive finding that positional encoding enhances learning of recurrent neural networks (RNNs). Positional encoding is a high-dimensional representation of time indices on input data. Most famously, positional encoding complements the capabilities of Transformer neural networks, which lack an inherent mechanism for representing the data order. By contrast, RNNs can encode the temporal information of data points on their own, rendering their use of positional encoding seemingly redundant/unnecessary. Nonetheless, investigations through synthetic benchmarks reveal an advantage of coupling positional encoding and RNNs, especially for handling a large vocabulary that yields low-frequency tokens. Further scrutinization unveils that these low-frequency tokens destabilizes the gradients of vanilla RNNs, and the positional encoding resolves this instability. These results shed a new light on the utility of positional encoding beyond its canonical role as a timekeeper for Transformers.

## 1 Introduction

Since their invention in 2017, Transformer neural networks (Vaswani et al., 2017) have taken over as the gold standard processor/generator of time series data, from the more traditional models in the form of recurrent neural networks (RNNs; Elman, 1990). One of the most striking differences between the two models is the way they represent the temporal information of the data (i.e., the order of individual data points, or *tokens*, in the time series). On the one hand, RNNs encode temporal information by updating their internal state based on the input observations as well as the previous state. On the other hand, Transformers per se do *not* have a mechanism to represent the temporal order of data points; consequently, they rely on an external "clock" called *positional encoding*.

Positional encoding is a high-dimensional representation of time indices on input data (Gehring et al., 2017). For instance, its most basic and popular implementation utilizes sinusoidal waves of various predefined frequencies (Vaswani et al., 2017). Positional encoding "timestamps" input tokens by addition/concatenation of these vectors to the corresponding input embeddings. Unlike RNNs, the time representation via positional encoding is invariant to input values (i.e., autonomous) until they are processed together by a network.

Although positional encoding has been considered as an *alternative* form of time representation that replaces RNNs (in combination with Transformers), positional encoding and RNNs are *not* fundamentally incompatible; that is, inputs to RNNs can be "redundantly" augmented by position-encoding vectors. Indeed, autonomous activities of biological neurons—such as neural oscillations—are thought to play an important role in time perception (Matell & Meck, 2004; Buhusi & Meck, 2005), as well as other perceptual processes (including vision, Milner, 1974; Eckhorn et al., 1988; Gray et al., 1989; and odors, Milner, 1974; Eckhorn et al., 1988; Gray et al., 1989; Wehr & Laurent, 1996; Perez-Orive et al., 2002) and motor control (Marder & Bucher, 2001; Gross et al., 2002; Proctor et al., 2010).

Accordingly, the present study explores "redundant" positional encoding on inputs to RNNs, utilizing synthetic benchmarks. The experiments will unveil that positional encoding aids RNNs in handling a wider variety of discrete inputs (i.e., a larger vocabulary) than those without positional encoding.

The contributions of this study are summarized as follows:

- Difficulties in training RNNs on a large vocabulary are demonstrated through systematically designed benchmark tasks. This problem has not been identified in the previous studies—or at most has received little attention from the research community—despite its potential relevance to empirical applications.

- This identified problem with training RNNs on a large vocabulary is elucidated by the gradient instability induced by low-frequency tokens, which necessarily arise from the expansion of the vocabulary size.

- A novel efficacy of positional encoding—besides its timestamping function for Transformers—is shown by coupling it with RNNs. Specifically, positional encoding will be shown to mitigate the large-vocabulary problem by stabilizing the gradients of RNNs against disruptions caused by low-frequency tokens.

## 2 Related Studies

### 2.1 Theoretical and Empirical Computational Power of (Vanilla) RNNs

Mathematically, RNNs are known to be Turing-complete; they can simulate Turing machines if their weights have infinite precision and are ideally tuned (Siegelmann & Sontag, 1992; 1994; 1995; Siegelmann, 1996; 1999; Chen et al., 2018).[1] Indeed, even RNNs with random recurrent and input-to-hidden weights (called *reservoir computers*; Maass et al., 2002; Jaeger & Haas, 2004) can achieve the universal approximation property if their hidden-to-output weights are idealized (Grigoryeva & Ortega, 2018; Gonon & Ortega, 2020). These theoretical results have motivated the use of RNNs for processing complex time series such as human languages (Sundermeyer et al., 2012; Graves, 2013) and weather (Shi et al., 2015).

In practice, however, RNN weights are bounded by finite precision and must be optimized based on finite observations of data. These settings impose limitations on the empirical capabilities of RNNs (Chen et al., 2018; Weiss et al., 2018). For example, empirical RNNs cannot store infinitely many observations in their memory, or state vector(s), and the memorized information decays over time. This latter problem with the memory duration has attracted the interest of researchers, leading to extensive exploration of RNN architectures for a longer-lasting memory (Hochreiter & Schmidhuber, 1997; Arjovsky et al., 2016; Neil et al., 2016; Chang et al., 2017; Jing et al., 2017; 2019).

More recently, research into prolonged memory retention has shifted towards continuous-time models (Voelker et al., 2019; Gu et al., 2020). Instead of representing the memory of an input sequence through discrete-time modifications of a latent state, these models approximate the input history by a linear combination of orthogonal polynomials in continuous-time space. Consequently, the coefficients of these polynomials yield a finite-dimensional representation of the input sequence (termed the High-Order Polynomial Projection Operator, or HiPPO) and the dynamics of these coefficients can be articulated through an ordinary differential equation (ODE; Gu et al., 2020). This concept of continuous-time memory representation has subsequently been extended to neural state-space models by replacing the fixed state matrix in the ODE—corresponding to a manually selected basis of polynomials—with a learnable one, while constraining its structure to a diagonal (plus a row-rank) matrix (Gu et al., 2021; 2022a;b; Gu & Dao, 2024). Most strikingly, with additional refinements, the latest state-space model has achieved superior language modeling performance, rivaling that of Transformer-based models.

---

[1]In order to be Turing-complete, RNNs must also be allowed to read an entire input prior to their output emission; they are at most context-sensitive if they have to return an output at each time step upon the receival of an input token (Chen et al., 2018; Weiss et al., 2018).

## 2.2 Positional Encoding

Positional encoding is a high-dimensional representation of temporal structures in input data (Gehring et al., 2017). The primary demand for this representation scheme comes from Transformers. Unlike RNNs, Transformers lack an inherent mechanism for representing input arrangements. Accordingly, input tokens to a Transformer are "time-stamped" via addition/concatenation of a position-encoding vector.

In the original implementation of Transformer, token positions were encoded by sinusoidal waves of various predefined frequencies (Vaswani et al., 2017). While this original encoding scheme is powerful enough for a wide range of tasks, researchers have explored for other possibilities as well. For example, the well-known BERT pretraining for natural language processing employed learnable embeddings to encode token positions (Devlin et al., 2019). Some studies also suggested that the combination of the sinusoidal and learnable encoding improves model performance (Dai et al., 2019). There is also an option to encode the *distance* between tokens rather than the elapsed time from the sequence onset (Shaw et al., 2018; Dai et al., 2019).

Besides Transformers, positional encoding is also used to represent the elapsed time in diffusion processes (Ho et al., 2020; Song et al., 2021). Moreover, the effectiveness of positional encoding is not limited to *temporal* information; previous studies in three-dimensional mesh/point-cloud modeling report that sinusoidal transformation of *spatial* data improves model performance compared to the raw coordinate representation (Mildenhall et al., 2020; Jun & Nichol, 2023).

Despite the widespread use of positional encoding in various areas of machine learning today, its applications to pure RNNs remain largely unexplored. To the best of the author's knowledge, only two studies have previously explored position-encoded RNNs.[2] Karanikolos & Refanidis (2019) reported that a position-encoded LSTM outperformed a vanilla LSTM as well as a shallow Transformer (with four layers) in text summarization. In another study—predating the proposal of sinusoidal positional encoding in the deep learning community—Vincent-Lamarre et al. (2016) showed that oscillatory signals at random frequencies improved the performance of a random RNN (i.e., reservoir computer) in a timing task, assessing the model's memory duration by its ability to produce a (smoothed) output pulse after a specified time interval from an onset signal.

Similarly, the time index in time series data has rarely been utilized directly by RNNs, presumably because of its "redundancy" along the functionality of RNNs. As an exception, Neil et al. (2016) proposed a periodic gating mechanism for updating the state and memory cell of LSTM. This periodic gating was scheduled according to a triangular wave interspersed with a plateau at the floor value (= 0.0; the frequency, phase, and the duration of the wave phase were learnable parameters).

# 3 Methods

## 3.1 Task

Effects of positional encoding on RNNs were investigated through the reverse-ordering task;[3] given a sequence of random integers, RNNs were trained to reconstruct them in reverse order (e.g. $8, 29, 2, 11 \mapsto 11, 2, 29, 8$; Fig. 1).

## 3.2 Model Architecture

The investigations in this study were based on single-layer gated recurrent unit (GRU; Cho et al., 2014) and long short-term memory (LSTM; Hochreiter & Schmidhuber, 1997) combined with input-embedding and output-projection layers (Fig. 1). Besides these RNNs, a neural state-space model, S4D (i.e., S4 with a diagonal state matrix; Gu et al., 2022a), was also investigated. Each of the integers in the input sequences was first embedded and concatenated with the positional encoding, and then fed to the RNN/S4D.[4] After

---

[2]In other studies, positional encoding and RNNs have served as submodules within more complex models, typically in conjunction with an attention mechanism (Kim et al., 2018; Song et al., 2020).

[3]Investigations of additional tasks are reported in Appendix A.

[4]The concatenation of the positional encoding and input embeddings inflates the number of learnable parameters in the input-to-hidden weights of RNNs in comparison to the vanilla models. However, this additional parameterization is designed

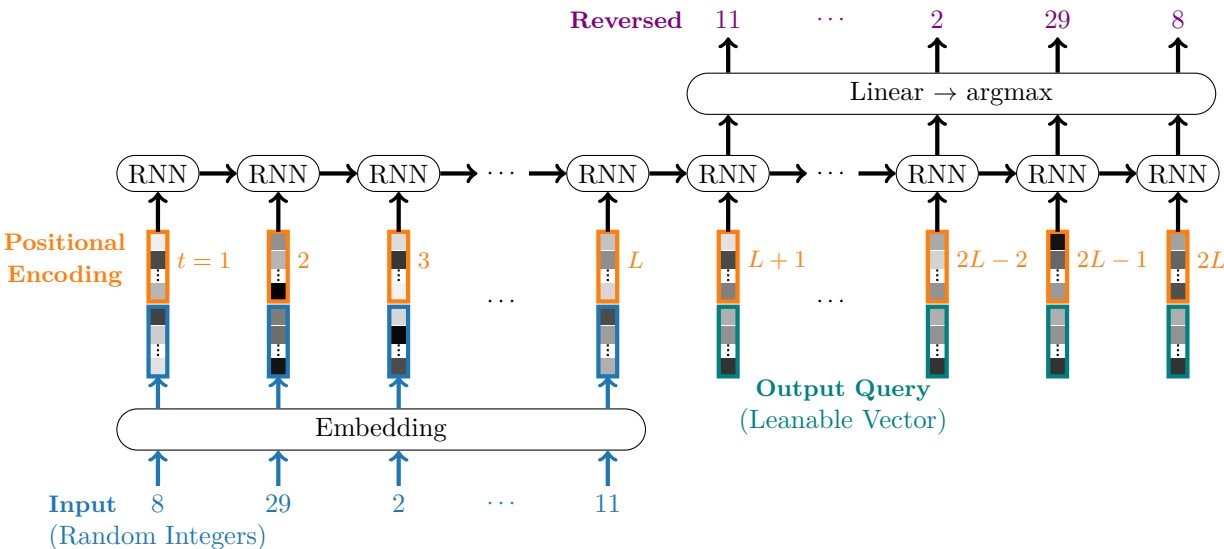

Figure 1: Illustration of the model structure and the reverse-ordering task.

reading the entire input sequence, the network received a command to return the output. This command was represented in the form of a time-invariant learnable vector, and fed to the RNN in place of the input embedding (cf. Arjovsky et al., 2016). The outputs from the RNN/S4D module were linearly projected into the classification logits, whose cross-entropy loss against the target sequence was used to optimize the entire network. Model predictions in the testing phase were defined by the argmax of these logits for each time step.

This study adopted the canonical sinusoidal positional encoding designed for Transformers (Vaswani et al., 2017, see Appendix D for discussions on alternative implementations); specifically, each time step $t$ was encoded by the $D_{pos}$-dimensional vector, $(PE_{t,1}, \ldots, PE_{t,D_{pos}})^T$, defined as follows:[5]

$$PE_{t,2i} := \sin\left(\frac{t-1}{10000^{\frac{2(i-1)}{D_{pos}}}}\right) \Big/ \sqrt{\frac{D_{pos}}{2}} \tag{1}$$

$$PE_{t,2i+1} := \cos\left(\frac{t-1}{10000^{\frac{2(i-1)}{D_{pos}}}}\right) \Big/ \sqrt{\frac{D_{pos}}{2}} \tag{2}$$

For the sake of learning stability, the positional encoding was divided by $\sqrt{D_{pos}/2}$ so that the encoding vectors had the unit L2-norm. Note that the time step $t$ increased throughout both the input and output phases (i.e., $t = 1, \ldots, L, L+1, \ldots, 2L$ where $L$ represents the input length), without any hard-coded association between the input and output positions (i.e., no shared timestamps).

### 3.3 Implementation Details

Across the experiments, the dimensionality of the hidden layer of the RNNs was set to 512. The embedding of the input integers and the memory cell of the LSTM also had the same dimensionality of 512. Similarly, the

---

not to impact the learning of input embeddings. Appendix C experimentally demonstrates that merely increasing the model size does not enhance the performance of RNNs.

[5]The adjustment of the time and frequency indices ("$t-1$" and "$i-1$") in Eqs. 1 and 2 aligns the 1-based indexing in this paper (adopted for better readability) with the 0-based indexing in Python.

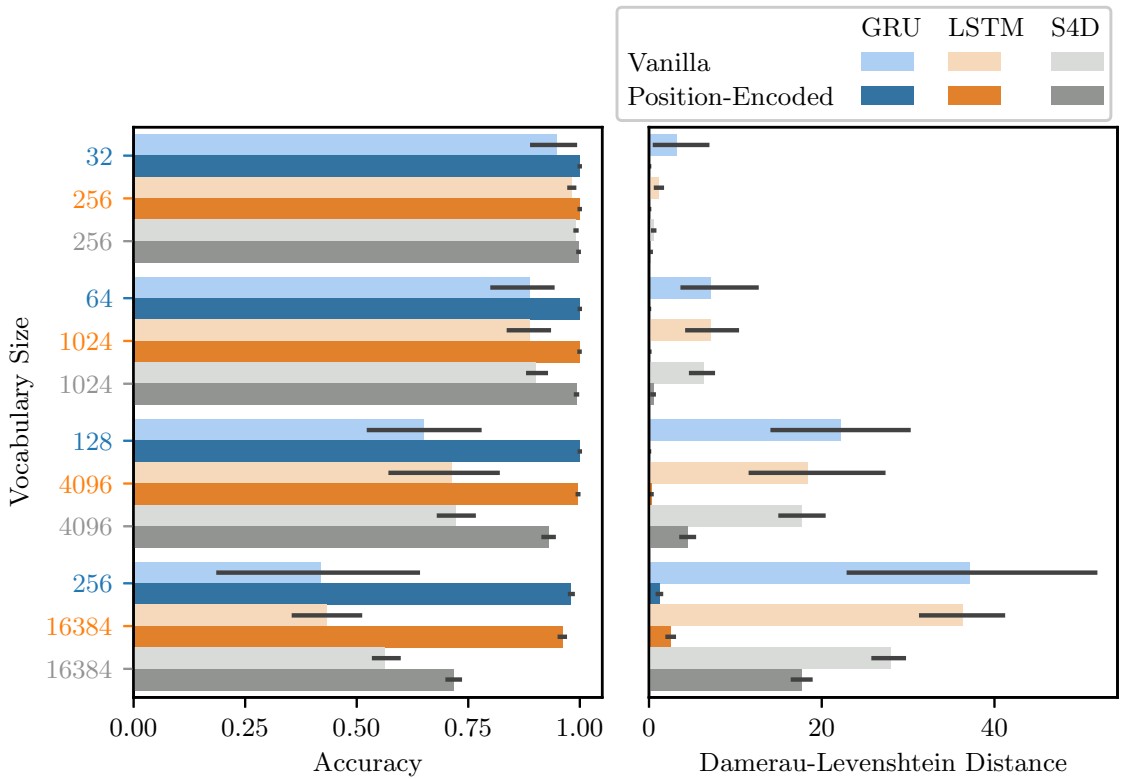

Figure 2: Token-wise accuracy (left) and sequence-wise reconstruction errors (right) of the reverse-ordering task performed by GRU/LSTM/S4D with and without positional encoding (labeled as "Position-Encoded" and "Vanilla" respectively). The input length was fixed at 64. The error bars represent the 95% confidence interval estimated from 10,000 bootstrapped samples of five training-test trials with different random seeds. Each of the five trials held out 1024 random sequences (= 65,536 tokens) for computing the test accuracy.

hidden dimensionality of S4D was set to 512, while its state size (or the order of the Legendre polynomials) was maintained at the default value of 64.[6]

The models were trained for 300,000 iterations using the Adam optimizer (Kingma & Ba, 2015) with the parameters $(\beta_1, \beta_2) := (0.9, 0.999)$ and no weight decay. The learning rate was linearly warmed up from 0.0 to 0.001 for the first 1,000 iterations, and then annealed according to the cosine schedule (Loshchilov & Hutter, 2017). The batch size was 512.

All the experiments were implemented in PyTorch (ver. 2.1.1; Paszke et al., 2017; 2019) and each training-test trial was executed on a single NVIDIA A100 GPU (with 80GB VRAM) hosted by the Academic Center for Computing and Media Studies, Kyoto University. The source code is available in https://github.com/tkc-morita/position-encoded_rnn.

# 4 Results

## 4.1 Key Findings

As a result, positional encoding improved the ability of RNNs to handle a larger vocabulary in the reverse-ordering task (Fig. 2). The position-encoded GRU and LSTM successfully reversed the input sequences of

---

[6]The S4D was implemented based on the official code provided in https://github.com/state-spaces/s4/blob/main/models/s4/s4d.py.

64 integers[7] drawn uniformly at random from the vocabularies of size 32–256 and 256–16,384, respectively, achieving the token-wise accuracy above 95%. By contrast, the performance of the vanilla models without positional encoding degraded as the vocabulary size increased. Similarly, positional encoding enhanced the capacity of S4D to handle large vocabularies. These improvements are also evident in the reduced sequence-wise reconstruction errors, measured by the Damerau-Levenshtein distance. It is of note that neither extra training iterations nor greater batch sizes improved the performance of the vanilla models.

## 4.2 Frequency Matters

The most apparent consequence of the increased vocabulary size was the reduced chance of observing individual vocabulary items (e.g., $1/256 \rightarrow 1/16{,}384$). Accordingly, additional experiments were conducted with non-uniformly distributed tokens to investigate the relation between their frequency vs. RNN performance. Specifically, the input vocabulary was evenly divided into FREQUENT and RARE groups, and the FREQUENT tokens had three times the probability of the RARE tokens; that is, the probability of each FREQUENT token was $7/8 \times 2/K$ (where $K$ denotes the total vocabulary size and was set to 64, 1024, 2048 for GRU, LSTM, and S4D respectively) whilst the probability of each RARE token was $1/8 \times 2/K$.

The training data consisted of 64 independent samples from this dual-frequency vocabulary. By contrast, the test data were systematically constructed so that each sequence included a single "target" token (FREQUENT/RARE) whose retrieval was evaluated for accuracy assessment, along with 63 "disturbants" that were either all FREQUENT or all RARE.

The experiment revealed that it was the *disturbant* tokens whose frequency significantly impacted the performance of the vanilla RNNs and S4D (Fig. 3). On the one hand, the RARE targets were successfully retrieved as long as they were surrounded by the FREQUENT disturbants. On the other hand, the vanilla GRU struggled to recover the FREQUENT targets when the other input tokens were filled with the RARE disturbants. The LSTM performance was also degraded, especially when the targets were positioned in the first quarter of the input sequence ($1 \leq t \leq 16$). Similarly, the RARE disturbants were detrimental to the S4D; unlike the RNNs, however, the accuracy was worst when the targets were located in the middle of the input sequences ($17 \leq t \leq 32$).

By contrast, the position-encoded RNNs exhibited robustness to the frequency of the target and disturbant tokens. They achieved nearly perfect accuracies in most cases, except when the GRU processed the fully RARE data whose target was located in the first half of the sequence ($1 \leq t \leq 32$). Likewise, positional encoding enhanced the resilience of the S4D against the influence of RARE disturbants.

## 4.3 Analysis of Gradient Stability

To delve deeper into the influence of token frequency on the RNN performance, the gradients of the RNN latent states were scrutinized. In the analysis, pairs of input sequences were processed by the RNNs trained on the dual-frequency vocabulary (comprising FREQUENT and RARE items; Fig. 4). Each pair of sequences shared the same initial token ($t = 1$; "target") but varied in the subsequent tokens ($2 \leq t \leq L$; "disturbants"). Then, gradients were computed for the distant mapping between the first and last updated states (i.e., at time $t = 1$ and $2L$) of the RNNs using backpropagation through time. The *stability* of RNN learning was assessed by measuring the dot-product similarity of the gradients between the paired input sequences (after normalization over output dimensions).

Formally, the paired input sequences, denoted as $A$ and $B$, established two distinct, but ideally similar mappings, $f^{(A)}$ and $f^{(B)}$, from the first to the last latent state of the RNNs ($\vec{\mathbf{h}}_{2L}^{(s)} = f^{(s)}(\vec{\mathbf{z}}_1)$, where $s \in \{A, B\}$). The gradient stability of the RNNs was defined by the dot-product similarities between the normalized gradients of these paired mappings:

---

[7]See Appendix B for a discussion of robustness to variable input length.

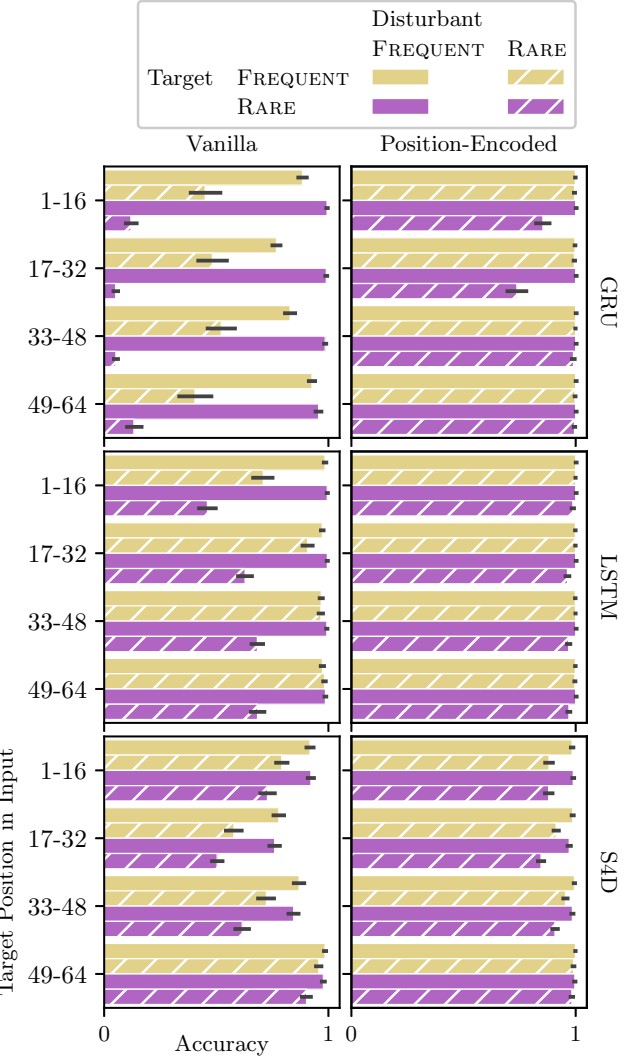

Figure 3: Token-wise accuracy of the reverse-ordering task performed by GRU/LSTM/S4D with and without positional encoding (labeled as "Position-Encoded" and "Vanilla" respectively). The vocabulary was evenly split into FREQUENT and RARE groups (32+32 for GRU, 512+512 for LSTM, and 1024+1024 for S4D), and the former was sampled three times more frequently than the latter. The input length was fixed at 64. The error bars represent the 95% confidence interval estimated from 10,000 bootstrapped samples of five training-test trials with different random seeds. Each of the five trials held out 4096 test sequences (= 262,144 tokens), consisting of a single "target" token (frequent or rare) surrounded by 63 "disturbants" (all frequent or all rare). That is, sixteen test sequences were held out for each condition (frequent/rare target × frequent/rare disturbants × target positions).

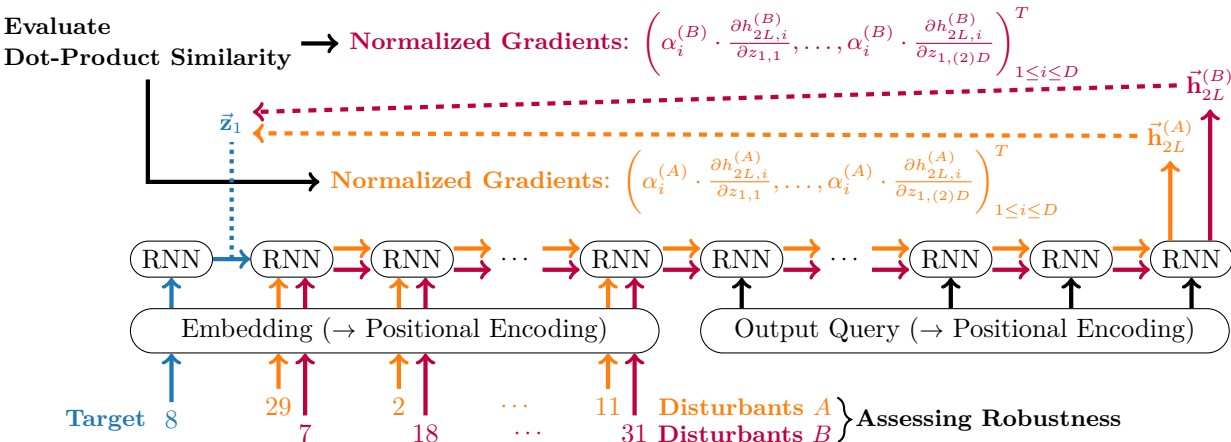

Figure 4: Schematic illustration of the analysis of gradient stability. The RNN trained on the reverse-ordering task processed a pair of input sequences that shared the initial token ($t = 1$; blue) but differed in the rest ($2 \leq t \leq L$; referred to as "Disturbant A/B", and colored in orange/purple). For each dimension $i$ of the final RNN output $h_{2L,i}$ at time $2L$, the distant gradient $\left(\frac{\partial h_{2L,i}}{\partial z_{1,1}}, \dots, \frac{\partial h_{2L,i}}{\partial z_{1,(2)D}}\right)^T$ at the first updated latent state $\vec{z}_1$ ($= \vec{h}_1$ in GRU; $=$ concatenation of the hidden and cell states in LSTM, doubling the total dimensionality to $2D$) was computed per input sequence via backpropagation through time (dashed lines). The gradient stability was defined by the dot-product similarity of the paired gradients normalized over the output dimensions by the coefficients $\alpha_i^{(s)}$ ($s \in \{A, B\}$), whose definition is provided in Eq. 4.

$$\text{STABILITY}(A, B) := \sum_{i=1}^{D} \langle \alpha_i^{(A)} \nabla f_i^{(A)}(\vec{z}_1), \alpha_i^{(B)} \nabla f_i^{(B)}(\vec{z}_1) \rangle = \sum_{i=1}^{D} \alpha_i^{(A)} \alpha_i^{(B)} \sum_{j=1}^{(2)D} \frac{\partial h_{2L,i}^{(A)}}{\partial z_{1,j}} \cdot \frac{\partial h_{2L,i}^{(B)}}{\partial z_{1,j}} \tag{3}$$

where the coefficients $\alpha_i^{(s)}$ normalized the raw gradients $\nabla f_i^{(s)}(\vec{z}_1)$ over the output dimensions $i := 1, \dots, D$ (i.e., over the row vectors of the Jacobian matrix):

$$\alpha_i^{(s)} := \frac{\|\nabla f_i^{(s)}(\vec{z}_1)\|}{\sum_{k=1}^{D} \|\nabla f_k^{(s)}(\vec{z}_1)\|} = \left(\sqrt{\sum_{j=1}^{(2)D} \left|\frac{\partial h_{2L,i}^{(s)}}{\partial z_{1,j}}\right|^2}\right) \Bigg/ \left(\sum_{k=1}^{D} \sqrt{\sum_{j=1}^{(2)D} \left|\frac{\partial h_{2L,k}^{(s)}}{\partial z_{1,j}}\right|^2}\right) \tag{4}$$

Consequently, the stability metric emphasizes the consistency of the paired gradients that *both* have a greater $L2$-norm across the output dimensions.

It deserves special note that the mapping from the first to the last RNN state was *conditioned on the disturbant tokens* occurring at $2 \leq t \leq L$. Nevertheless, the reverse-ordering task trained the networks to retrieve the initial token as their final output whatever tokens intervened in the middle. Thus, a well-trained RNN would maintain invariance in its final state over the disturbants. Conversely, consistent gradient directions across varied disturbants would lead to successful learning, which premises the proposed analysis.

Unlike the RNN models, both the vanilla and position-encoded S4Ds achieved high accuracy over 96% for the initial target token ($t = 1$), regardless of the frequency of the target and disturbants. Accordingly, for the analysis of S4D, the target token was positioned in the middle at $t = 23$, where the vanilla model exhibited its poorest accuracy with the RARE disturbants. The disturbants were prefixed and suffixed to this target to construct input sequences. The prefix disturbants were shared between the paired sequences, thereby ensuring that the latent dynamics of the model remained identical up to the target token.

It should also be noted that the latent states of S4D are complex-valued (while its outputs are real-valued), and consequently, the gradients and their dot-product similarities are also complex-valued. For the sake of

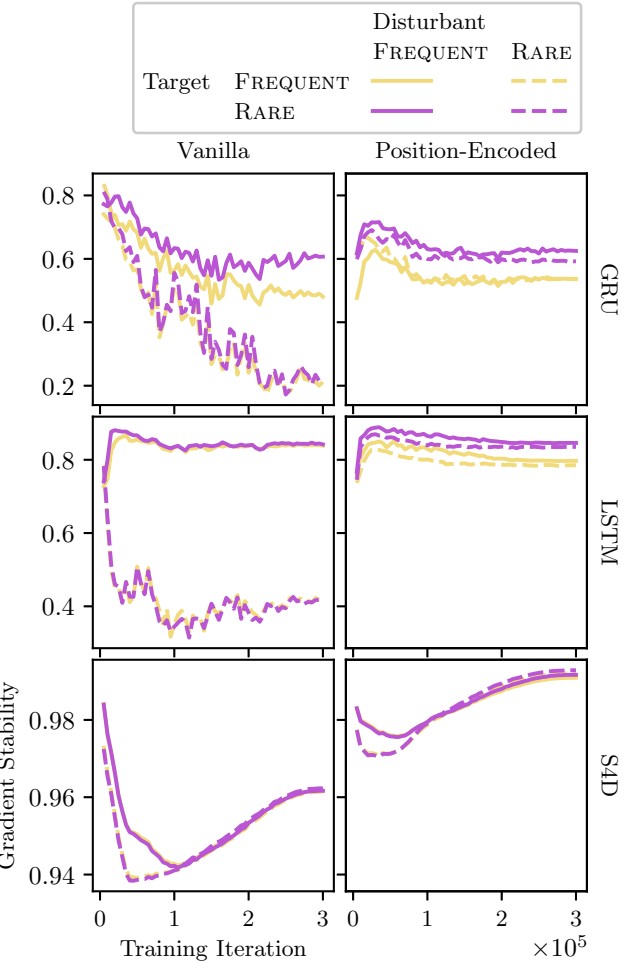

Figure 5: Gradient stability of GRU/LSTM/S4D trained on the reverse-ordering task with and without positional encoding (labeled as "Position-Encoded" and "Vanilla" respectively). For the GRU and LSTM, the stability was defined by the dot-product similarity of latent-to-latent gradients after normalization over the output dimensions, conditioned on two input sequences sharing the initial "target" token (whose FREQUENT vs. RARE distinction is represented by the line color), followed by FREQUENT or RARE disturbants (represented by the solid vs. dashed lines). For the S4D, the target token was positioned at $t = 23$, where the vanilla model scored the worst accuracy with the RARE disturbants. The disturbants were prefixed and suffixed to the target to construct input sequences. The prefix disturbants were shared between the paired sequences, ensuring that the latent dynamics of the model was guaranteed to remain identical up to the target token. The total input length was $1 + 63 = 22 + 1 + 41 = 64$. The average similarity over 1024 input pairs times five trials is plotted for every 5000 training iterations.

the present analysis, the complex-valued gradients were treated as the double-sized real arrays, and a real-valued similarity was defined by Eq. 3. This is equivalent to taking the real component of the complex-valued similarity, and is intuitively natural given that a perfect alignment between complex gradient directions yields a real-valued score of 1.0 (= the norm of a normalized complex vector). Additionally, the extra dimension in the latent states representing the order of the Legendre polynomials was merged with the channel dimension, and the entire state was treated as a flattened vector.

Monitoring the gradients at training checkpoints revealed that RARE disturbants destabilize the learning of vanilla RNNs (Fig. 5). The similarity of the paired gradients decreased gradually (GRU) or rapidly (LSTM) when the networks were exposed to the RARE disturbants.

And most remarkably, positional encoding endowed the RNNs with robustness to these RARE disturbants. Both the GRU and LSTM maintained the high similarity of the paired gradients across the different target/disturbant conditions.

By contrast, the impact of positional encoding on the gradient stability of the S4D was marginal; unlike the RNNs, the vanilla S4D was highly stable by itself against RARE disturbants throughout the training, even though there was a visible relative destabilization due to RARE disturbants compared to FREQUENT disturbants in the early stages of training, as well as an observable improvement by positional encoding. It is also noteworthy that the difference between the FREQUENT vs. RARE disturbants diminished after 10,000 training iterations. Consequently, gradient stability does not fully account for the decline in the accuracy of S4D in the presence of RARE disturbants, nor does it explain the enhancement brought about by positional encoding.

## 5 Discussion

### 5.1 Difficulties in Handling a Large Vocabulary

The present study introduced a novel challenge in training (vanilla) RNNs: large vocabularies. While investigations into the manageable vocabulary size of RNNs appear to be a pertinent research area—being crucial for empirical applications such as natural language processing—previous studies were primarily dedicated to evaluating and improving the *memory duration* of RNNs, and the vocabulary size in these studies was typically set small (= eight; Arjovsky et al., 2016; Neil et al., 2016; Chang et al., 2017; Jing et al., 2017; 2019; Voelker et al., 2019; Gu et al., 2020).

This study examined the RNN gradients and identified their destabilization when processing low-frequency tokens, which are necessarily included in a large vocabulary. Specifically, inputs that do *not* contribute to gradient-based optimization at a target time step (e.g., tokens at $2 \leq t \leq L$ upon the retrieval of the initial token at $t = 2L$ in the reverse-ordering task) were found to be detrimental.

In general cases of time series processing, data points carrying crucial information for specific time steps become irrelevant otherwise. Consequently, each token exhibits a dual nature—both crucial and noisy—throughout the task, and processing rare tokens is particularly challenging presumably because they are irrelevant at most of the time while making a large impact on the learning through the greater loss to compensate for their fewer learning opportunities. Dealing with such "unignorable noise" presents a pervasive challenge for RNNs.

### 5.2 Functionality of Positional Encoding beyond the Timekeeper for Transformers

Although low-frequency tokens destabilize the gradient-based learning of RNNs, the present study also discovered that this issue can be alleviated by positional encoding. This enhancement of RNNs via positional encoding is noteworthy because RNNs were specifically designed to process time series data on their own; hence, unlike Transformers, they are presumed to function without relying on an "external clock" (Siegelmann & Sontag, 1992; 1994; 1995; Siegelmann, 1996; 1999). Consequently, position-encoded RNNs have remained largely unexplored, with only two exceptions to the best of the author's knowledge (Vincent-Lamarre et al., 2016; Karanikolos & Refanidis, 2019). The findings of the present study—namely, the improvement in the manageable vocabulary size due to the enhanced gradient stability—broaden the currently limited understanding of the impact of positional encoding on RNNs.

Additionally, the results of this study shed a new light on the utility of positional encoding. While positional encoding has been viewed as nothing more than input timestamps for Transformers, the present study demonstrated its efficacy in stabilizing the gradients of RNNs against disruption by low-frequency tokens. This novel functionality of positional encoding would not have been visible in Transformer studies, as the

model can dynamically adjust the relevance of input tokens through their attention mechanism and thus inherently mitigate the impact of disturbant tokens.

## 5.3 Limitations and Future Directions

A primary unresolved question in this study pertains to the mechanism behind the gradient stabilization by positional encoding. All the findings here are based on experimental investigations, lacking rigorous mathematical explanations for how and why the gradients of RNNs are destabilized by infrequent tokens and stabilized by positional encoding. Moreover, the present study primarily focused on the canonical implementation of sinusoidal positional encoding designed for Transformers (Eqs. 1,2), leaving it open which parameters of the sinusoidal waves (i.e., frequencies and phases) are critical for gradient stabilization. Future research may broaden its scope to encompass more general forms of positional encoding, such as wavelets and non-periodic signals (see Appendix D for a preliminary comparison among sinusoidal, learnable, and randomly-fixed encodings).

Moreover, the analysis of gradient stability did not fully address the enhanced performance of the position-encoded state-space model (S4D). In terms of accuracy, the positioned-encoded S4D exhibited greater robustness to infrequent tokens compared to the vanilla model, resembling the behavior observed in RNNs. However, the gradients of the vanilla S4D were too stable to account for this descline in performance. This leaves the question open how positional encoding influences gradient-based learning of state-space models. Additionally, future studies may investigate a broader range of state-space models—including the state-of-the-art architecture of Mamba (Gu & Dao, 2024)—to achieve a comprehensive understanding of the interplay between positional encoding and these models.

In addition to these scientifically oriented questions, future studies could also address practical applications of position-encoded RNNs and neural state-space models. Although positional encoding enhanced model performance across different synthetic tasks (see Appendix A), the extent of this enhancement is task-dependent. Indeed, while Karanikolos & Refanidis (2019) reported the effectiveness of positional encoding for an LSTM text summarizer, the present study found no empirical advantage for the language modeling task, aside from a slightly more rapid decline in training loss (see Appendix E). Thus, positional encoding is not a panacea for arbitrary tasks, and further investigations are necessarily to determine when it is effective.

## Acknowledgments

This study was supported by JST ACT-X (JPMJAX21AN) and Core Research for Evolutional Science and Technology (JPMJCR17A4, JPMJCR22P5); JSPS Grant-in-Aid for Early-Career Scientists (JP21K17805) and for Scientific Research A (JP24H00774), B (JP22H03914), and C (JP24K15087); and Kayamori Foundation of Informational Science Advancement (K35XXVIII620). The author also gratefully acknowledges the support of the ACCMS, Kyoto University, regarding the use of their supercomputer system.

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

## A    Other Tasks

This section demonstrates the effectiveness of positional encoding on RNNs across different tasks, besides the reverse ordering task discussed in the main text.

### A.1    Reverse-Ordering + Delayed-Addition

This section reports the performance of position-encoded RNNs on a more complicated, combinatorial task than the reverse ordering of input sequences. Extending the reverse-ordering task, the models received additional random input integers during the output phase, and added each of them to the corresponding token in the reverse-ordered input sequence (modulo the vocabulary size, so that the output range was bounded; Fig. 6).

This task was too challenging to GRUs—even after reducing the input length to $L = 16$—so only the results from LSTMs are reported below. Also, the network was trained for 600,000 iterations (i.e., twice longer

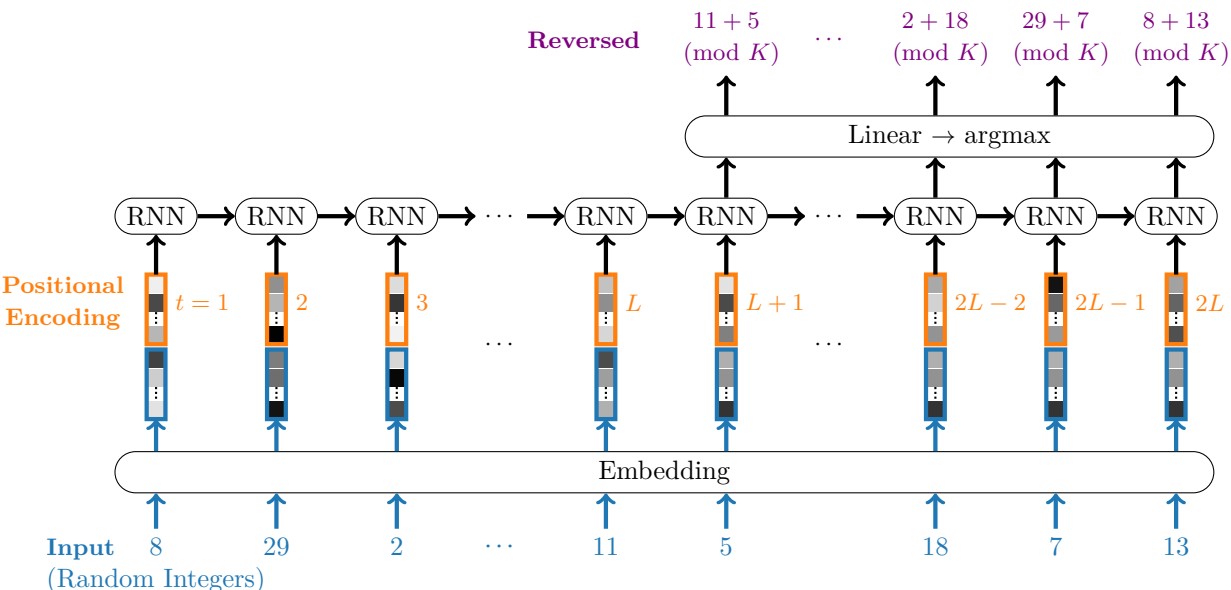

Figure 6: Illustration of the reverse-ordering + delayed-addition task. The modulus $K$ of the addition is equal to the vocabulary size.

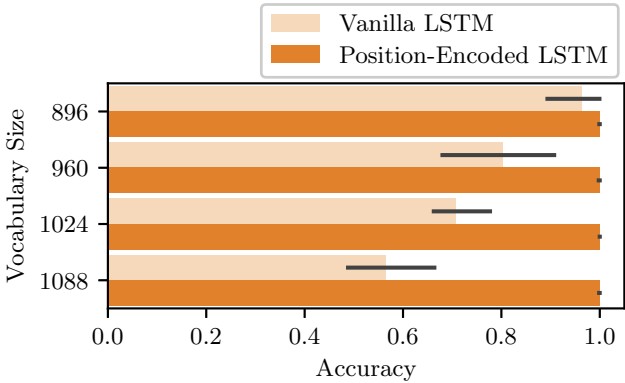

Figure 7: Token-wise accuracy of the reverse-ordering + delayed-addition task performed by the LSTM with and without positional encoding (labeled as "Position-Encoded" and "Vanilla" respectively). The input length was fixed at $L := 16$. The error bars represent the 95% confidence interval estimated from 10,000 bootstrapped samples of five training-test trials with different random seeds. Each of the five trials held out 1024 random sequences (= 16,384 tokens) for computing the test accuracy.

than the other tasks) for ensuring the convergence. The other conditions/hyperparameters were the same as reported in the main text.

Consequently, positional encoding improved the model performance as the vocabulary size grew from 896 to 1088 (Fig. 7).

## A.2 Sorting

In the reverse ordering task, the order of input integers was important information for accomplishing the task. Thus, positional encoding may play its originally intended role in encoding the temporal information.

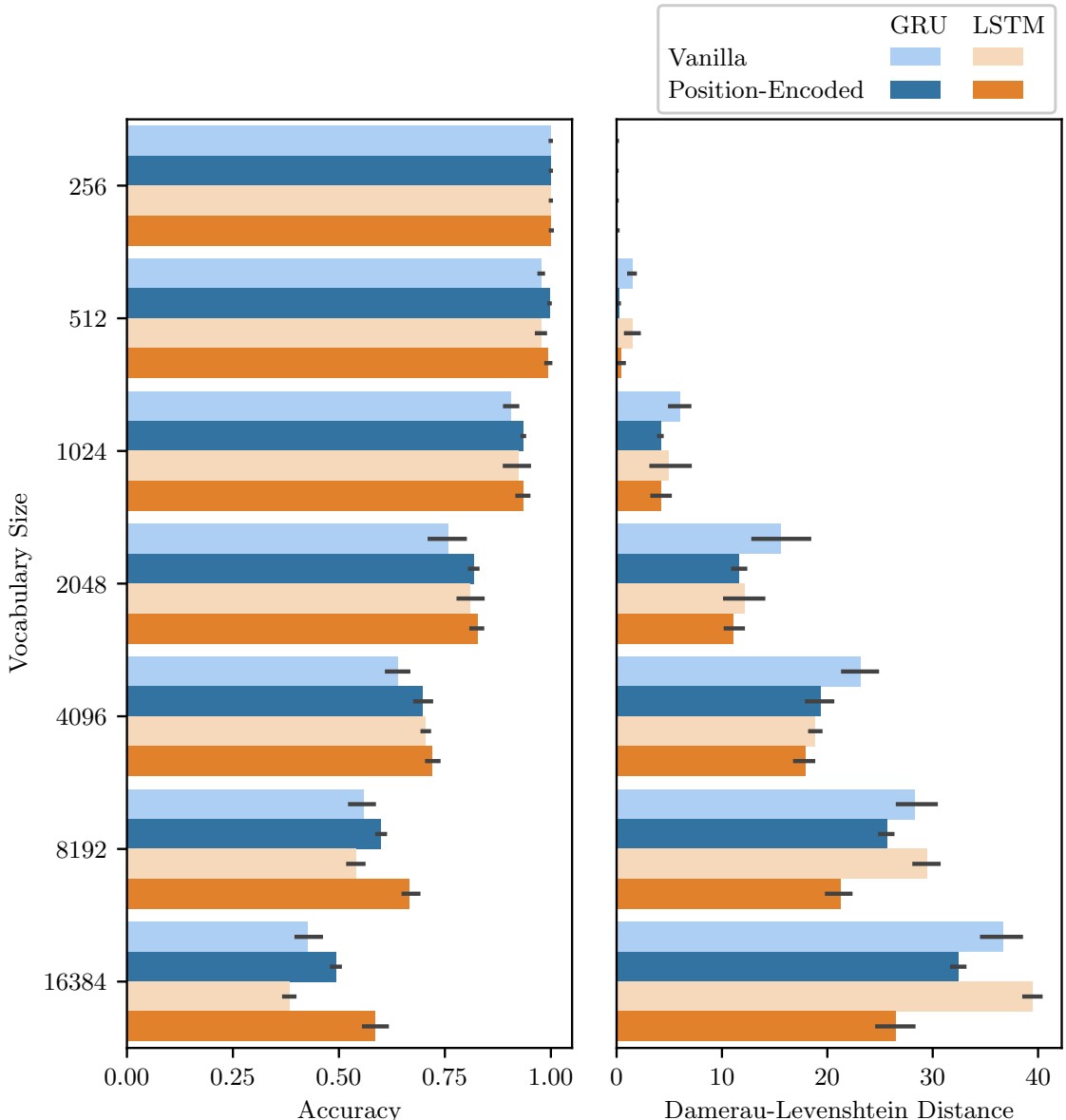

Figure 8: Token-wise accuracy (left) and sequence-wise reconstruction errors (right) of the sorting task performed by RNNs with and without positional encoding (labeled as "Position-Encoded" and "Vanilla" respectively). The input length was fixed at $L := 64$. The error bars represent the 95% confidence interval estimated from 10,000 bootstrapped samples of five training-test trials with different random seeds. Each of the five trials held out 1024 random sequences (= 65,536 tokens) for computing the test accuracy.

This section reports the effectiveness of positional encoding for a task in which the order of input observations was completely irrelevant; the learning objective was to simply sort the input integers in their inherent ascending order (e.g. $8, 29, 2, 11 \mapsto 2, 8, 11, 29$). The input integers were uniformly randomly sampled *with* replacement, allowing for ties in the sorting process.

As a result, positional encoding also proved effective for RNNs to handle a larger vocabulary in the sorting task (Fig. 8), though the improvement remained marginal compared to the reverse-ordering task.

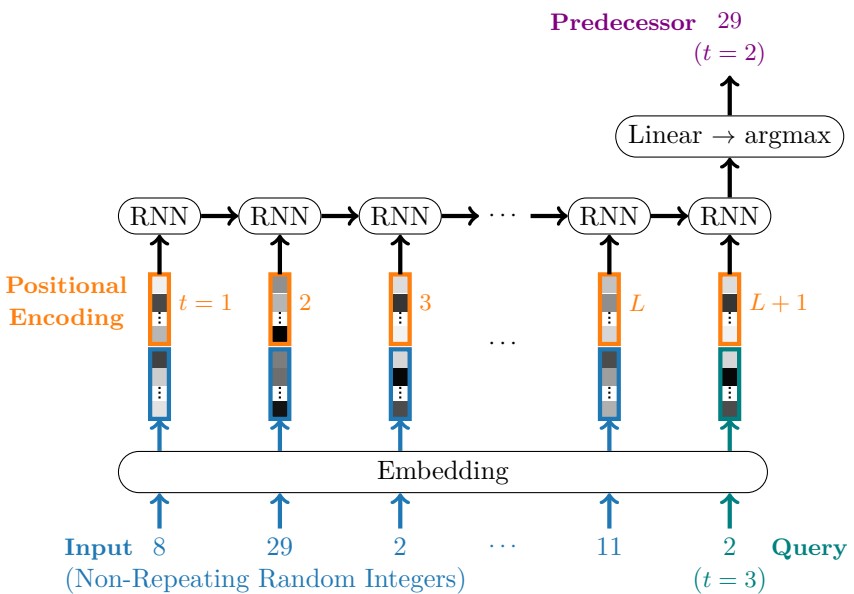

Figure 9: Illustration of the precedent-query task.

## A.3 Predecessor Query

Finally, this section presents benchmark results for the predecessor-query task, illustrated in Fig. 9. The network first received a sequence of *non-repeating* random integers, $x_1, \ldots, x_L$. Subsequently, one of the non-initial input integers, $x_{t_{\text{query}}}(2 \leq t_{\text{query}} \leq L)$, was randomly selected and reintroduced to the network at time $t = L+1$. The learning objective is to return the *predecessor* of the reviewed integer $(= x_{t_{\text{query}}-1})$. The predecessor-query task evaluates the capacity of RNNs to integrate information regarding both the order and content of input sequences.

As in the reverse-ordering + delayed-addition task, the input sequence was reduced to $L = 16$ due to the complexity of the task, and the experiment focused on the LSTM. The number of training iterations was maintained at 300,000.

Similar to the other benchmarks, positional encoding improved the LSTM's capacity to manage the larger vocabularies.

## B Robustness to Variations in Input Length

So far, all the tasks were experimented using fixed-length inputs ($L = 64$). One might wonder if positional encoding is *exceptionally* effective under this setting, informing RNNs with the exact timing when each input token should be returned as the output. Thus, it remains unclear whether or not position-encoded RNNs can also handle a larger vocabulary even when the input length is variable and, thus, the exact timing of the output emission is *not* identifiable from the positional encoding attached to the inputs.

To assess the robustness to variations in the input length, an additional experiment was conducted on the LSTM, with the input length varied between 32 and 64. In this setup, the maximum input length ($= 64$) covers the entirety of the shortest input sequence plus its reversed reconstruction ($= 32 + 32$). Consequently, the positional encoding per se cannot even distinguish the input vs. output phases at $t = 33, \ldots, 64$. The vocabulary size was set to 16,384.

As a result, the positional encoding still improved the LSTM's performance on the reverse-ordering task against the perturbations in the input length (Fig. 11). This result suggests that the effectiveness of the positional encoding for RNNs is not limited to strictly scheduled tasks.

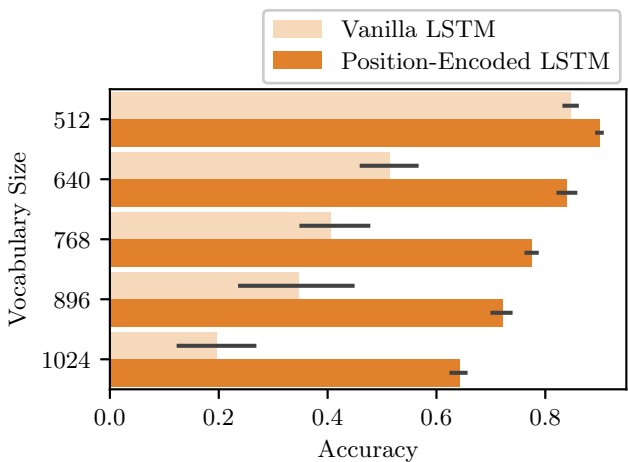

Figure 10: Accuracy of the precedent-query task performed by the LSTM with and without positional encoding (labeled as "Position-Encoded" and "Vanilla" respectively). The input length was fixed at $L := 16$. The error bars represent the 95% confidence interval estimated from 10,000 bootstrapped samples of five training-test trials with different random seeds. Each of the five trials held out 1024 random sequences (= 65,536 tokens) for computing the test accuracy.

## C   Effects of Additional Parameters in Position-Encoded RNNs

The concatenation of positional encoding with input embeddings inflates the number of learnable parameters in the input-to-hidden projection weights. This additional parameterization per se does not influence the learning of the input embeddings, and therefore does not elucidate the enhanced performance of position-encoded RNNs. This section substantiates this argument by equalizing the number of learnable parameters between the vanilla and position-encoded models.

Specifically, the equalization was achieved by concatenating two identical copies of the input embeddings and feeding them to the LSTM. This configuration—henceforth termed "double vanilla"—effectively doubled the size of the input-to-hidden weight for each gate in the LSTM, aligning it with that of the position-encoded LSTM, while maintaining all other parameters, including the dimensionality of the (non-repeated) input embeddings.

As illustrated in Fig. 12, the double vanilla LSTM did not yield any improvements in the reverse-ordering or sorting tasks. These results affirm that the reported enhancement of RNNs is not merely attributable to the additional parameterization associated with the positional encoding.

## D   Alternative Implementations of Positional Encoding

While this study implemented positional encoding by sinusoidal waves (Vaswani et al., 2017), there are alternative implementations proposed in the previous studies. For instance, the BERT-based models typically encode each token position by a learnable embedding (Devlin et al., 2019). Moreover, the original study of Transformer pointed out that even random vectors can function as positional encoding (Vaswani et al., 2017).

Accordingly, these two alternative forms of positional encoding were tested on the LSTM performing the reverse-ordering task. The random position-encoding vectors were uniformly and independently sampled from the $(512-1)$-dimensional hypersphere. The learnable embeddings were implemented using the canonical embedding module of PyTorch (`torch.nn.Embedding`). The input length and vocabulary size were set to 64 and 16,384 respectively. As shown in Fig. 13, both the random vectors and learnable embeddings improved the performance of LSTM.

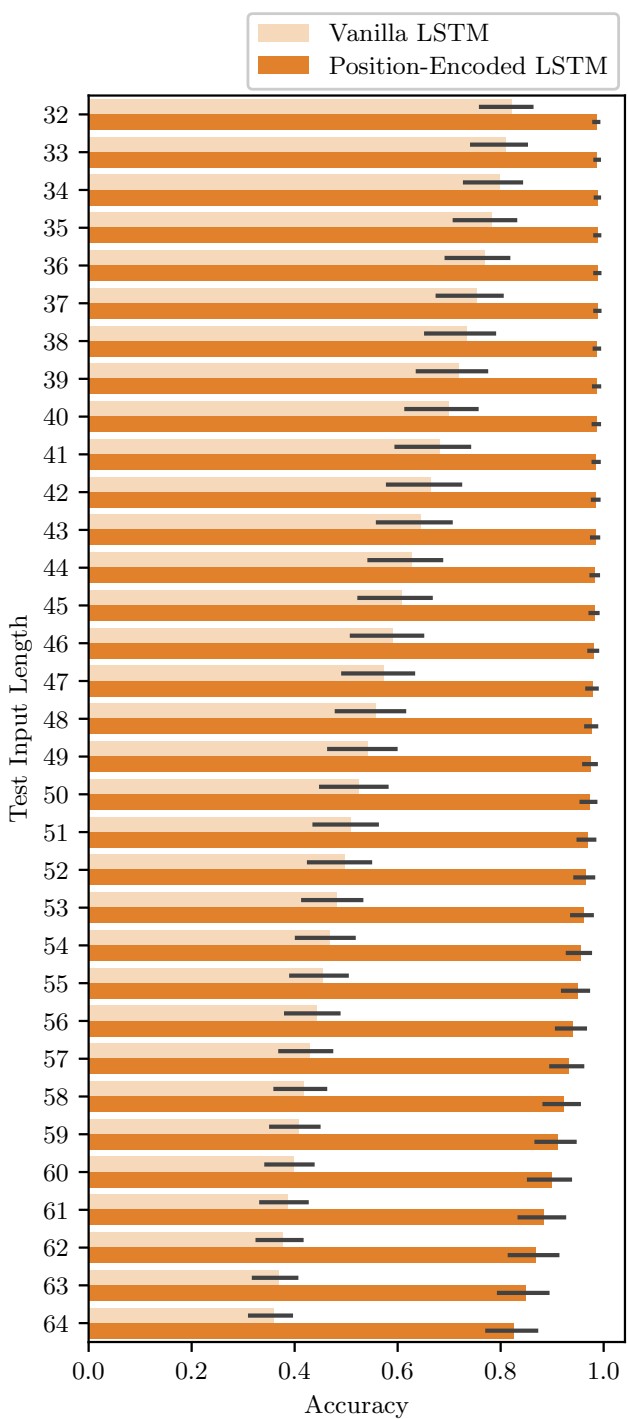

Figure 11: Token-wise accuracy of the reverse-ordering task performed by the LSTM with and without positional encoding (labeled as "Position-Encoded" and "Vanilla" respectively). During training, the input length was randomly selected from a range of 32 to 64. The vocabulary size was set to 16,384. The error bars represent the 95% confidence interval estimated from 10,000 bootstrapped samples of five training-test trials with different random seeds. Each of the five trials held out 1024 random sequences per length for computing the test accuracy.

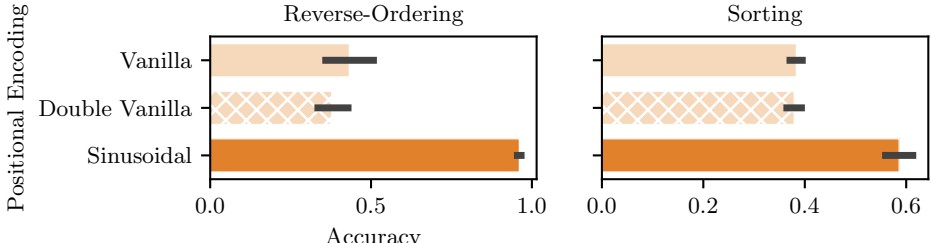

Figure 12: Token-wise accuracy of the reverse-ordering and sorting tasks performed by the LSTM with and without positional encoding. The "double vanilla" model concatenated two copies of the identical input embeddings to match the number of parameters with the position-encoded model. The input length was fixed at $L := 64$. The vocabulary size was set to 16,384. The error bars represent the 95% confidence interval estimated from 10,000 bootstrapped samples of five training-test trials with different random seeds. Each of the five trials held out 1024 random sequences (= 65,536 tokens) for computing the test accuracy.

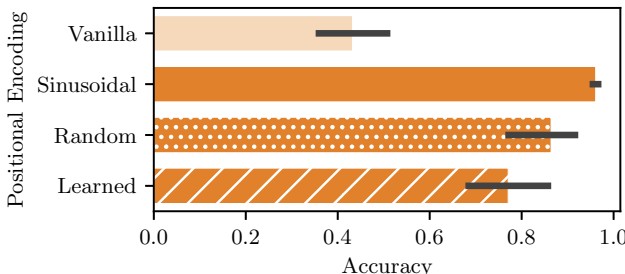

Figure 13: Token-wise accuracy of the reverse-ordering task performed by the LSTM with and without positional encoding. Three variants of positional encoding (sinusoidal, randomly fixed vectors, and learnable embeddings) were tested. The input length was fixed at $L := 64$. The vocabulary size was set to 16,384. The error bars represent the 95% confidence interval estimated from 10,000 bootstrapped samples of five training-test trials with different random seeds. Each of the five trials held out 1024 random sequences (= 65,536 tokens) for computing the test accuracy.

Among the different implementations of positional encoding, the sinusoidal encoding outperformed the two alternatives. The advantage of the sinusoidal encoding became more apparent when the input length was variable between 32 and 64 (Fig. 14); the sinusoidal encoding was more robust to the variations in the input length than the others.

## E    Language Modeling

This section reports benchmark results for the language modeling task. Single-layer LSTMs with and without sinusoidal positional encoding were trained and tested on the WikiText-103 dataset (Merity et al., 2017). Due to constraints in computational resources, the vocabulary was reduced from the original size of 267,735 to 32,768 by retokenizing the raw data using SentencePiece (Kudo & Richardson, 2018). The headings were removed, and the main text was segmented by paragraphs (separated by the line break). Additionally, only the first 1024 tokens of each paragraph were utilized for training and testing, ensuring that the absolute positional encoding always aligned with the beginning of each paragraph. The hyperparameters were configured as specified in §3.3.

As illustrated in Fig. 15, positional encoding proved effective only for marginally faster learning during the initial phase of training. The difference diminished around 10,000/30,000 iterations, and the test perplexities of the position-encoded model were inferior to those of the vanilla model.

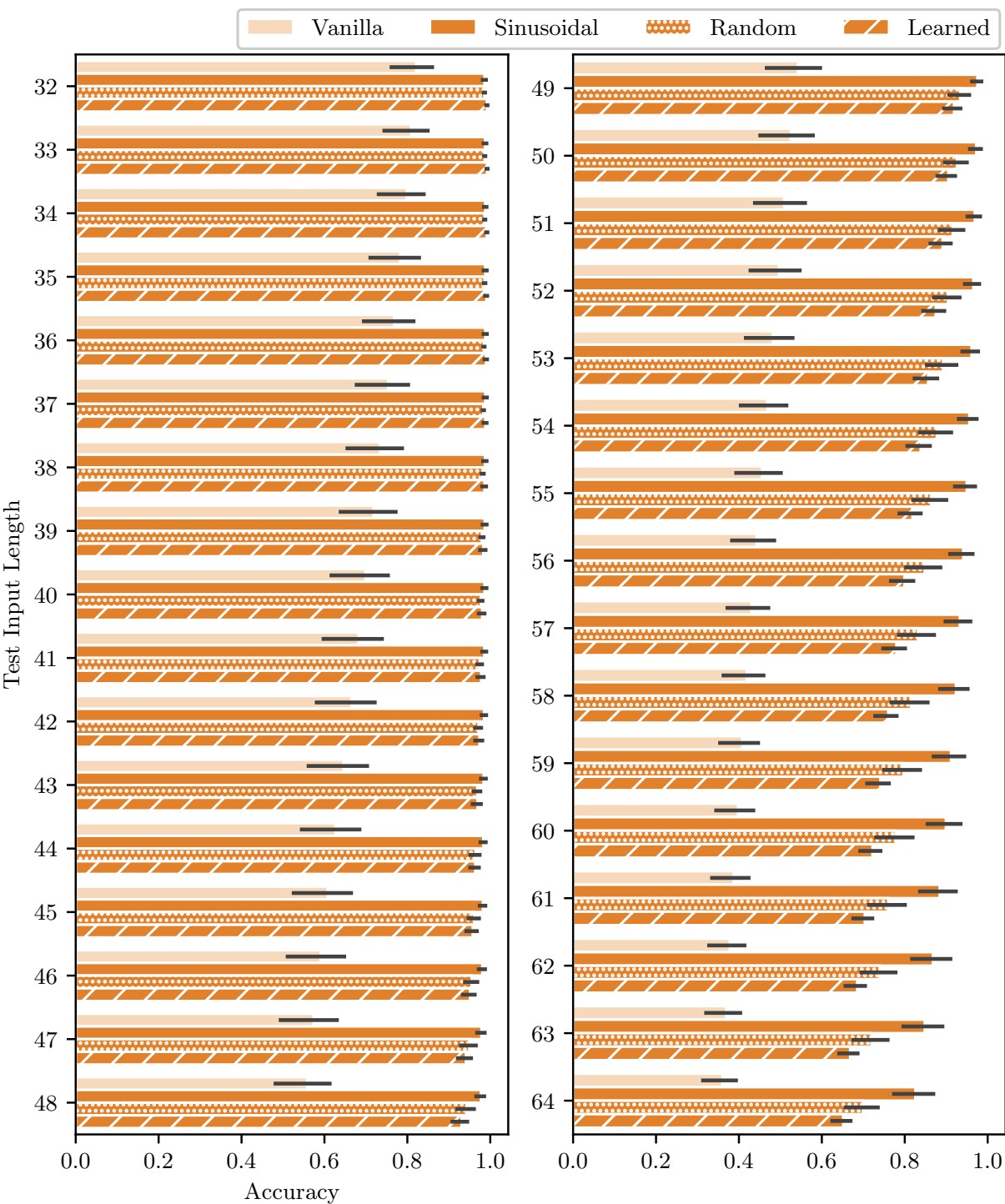

Figure 14: Token-wise accuracy of the reverse-ordering task performed by the LSTM with and without positional encoding. Three variants of positional encoding (sinusoidal, randomly fixed vectors, and learnable embeddings) were tested. During training, the input length was randomly selected from a range of 32 to 64. The vocabulary size was set to 16,384. The error bars represent the 95% confidence interval estimated from 10,000 bootstrapped samples of five training-test trials with different random seeds. Each of the five trials held out 1024 random sequences per length for computing the test accuracy.

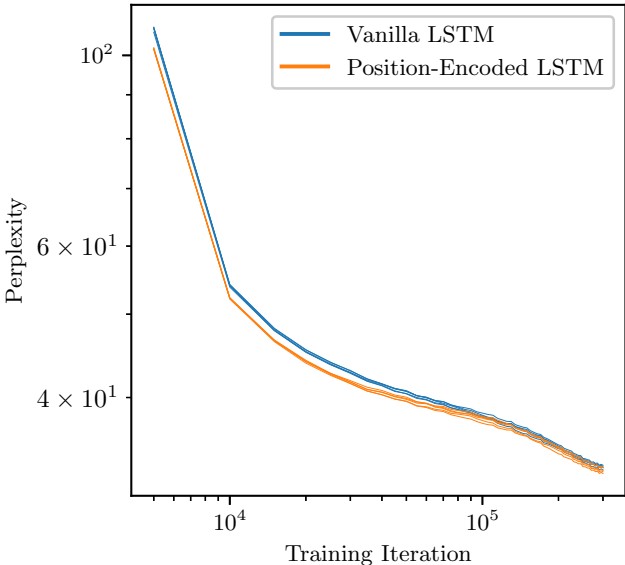

Figure 15: Training perplexities on the WikiText-103 dataset. Each of the five trials with a different random seed is represented by a single line.

Table 1: Test perplexities on the WikiText-103 dataset. The minimum, mean, and maximum are obtained from five trials with different random seeds.

| Model | Min | Mean | Max |
|---|---|---|---|
| Vanilla LSTM | **36.8257** | **37.7731** | 38.916589 |
| Position-Encoded LSTM | 38.0685 | 38.5384 | **38.893656** |

