# OpenReview forum: "Positional Encoding Helps Recurrent Neural Networks Handle a Large Vocabulary"
_TMLR — Accepted by TMLR_

### Review · Reviewer_7qMR · 2024-09-11

**Summary Of Contributions:**

This study investigates the effects of positional encoding on recurrent neural networks (RNNs). The authors report an interesting and somewhat counterintuitive finding: that positional encoding, typically associated with Transformer models, can also benefit RNNs, particularly when dealing with large vocabularies. The core argument is that low-frequency tokens in large vocabularies lead to gradient instability during RNN training, and positional encoding helps mitigate this issue.

**Audience:**

Yes

**Claims And Evidence:**

Yes

**Requested Changes:**

The only thing I would really like to see is more tasks and experiments than just relying on one single toyish task. Otherwise the paper brings in nice ideas.

**Strengths And Weaknesses:**

Strengths:

1) Novel Perspective: The paper challenges the conventional view that positional encoding is exclusive to Transformers. It opens up a new avenue for improving RNN performance, which could be valuable for tasks involving large and sparse vocabularies.
2)Clear Empirical Results: The experiments on synthetic benchmarks convincingly demonstrate the positive impact of positional encoding on RNNs, especially as the vocabulary size increases. The analysis of gradient stability provides further evidence supporting the authors' claims.
3) Well-Written: The paper is generally well-written and organized, making the ideas and findings accessible.

Weaknesses:

1) Limited Task Scope: The primary weakness lies in the reliance on a single task (reverse ordering) for the majority of the experiments. While the findings are compelling, their generalizability to other tasks remains an open question. Additional experiments on diverse tasks would significantly strengthen the paper's conclusions.
2) Lack of Real-World Application: Although the synthetic benchmarks provide a controlled environment for studying the phenomenon, the paper would benefit from demonstrating the practical implications of the findings on real-world tasks, such as natural language processing or time series prediction.
3) Limited Exploration of Alternative Encodings: The paper briefly touches upon alternative positional encoding schemes but doesn't explore them in depth. A more thorough comparison of different encodings could provide insights into their relative merits and limitations.

---

> ### Author Response · Authors · 2024-09-30
> **Authors' Response to Reviewer 7qMR**
>
> Thank you for your constructive feedback. Please find our responses below.
>
> ## Response to the Requested Changes
>
> An extra synthetic benchmark and a real-world benchmark will be added.
>
> The new synthetic benchmark evaluates models' ability to control their output depending on different queries. Specifically,
> 1. The models first receieve a squence of random integers WITHOUT DUPLICATIONS (i.e., no integer occurs more than once in a single sequence).
> 2. Then, the models is fed one of the observed numbers, $x_t$, randomly selected in the time range $2 \leq t \leq L$ where $L$ is the length of the sequence (i.e., non-initial observations).
> 3. The target output is $x_{t-1}$, or the token that immediately precedes the reviewed number in the input sequence.
>
> Unlike the reordering task, the models cannot prepare their output until they observe the ouput query at time $L+1$.
> The task also requires an ability to combine the knowledge about observed values (semantics) and order (syntax), simplifying a question answering like "John walked, Mary swimmed, and Alex drove. Q: Who walked?".
>
> As a result, similar improvements by positional encoding were observed, though the task was more challenging than the reverse ordering and even the position-encoded LSTMs failed to achieve a high accuracy (e.g., 64.30% by position-encoded vs. 19.59% by vanilla, on average among five trials).
>
> For the real-world benchmark, we are testing language modeling on wikitext-103 (w/ custom tokenization by SentencePiece, due to the limitations of computational resources). The task was selected simply because it is the most canonical in this era of LLM. At the time of this post, four trials (with different random seeds) are done for vanilla and position-encoded LSTMs, and so far,position-encoded LSTMs seem to learn a bit faster (in terms of the drop in training perplexities) especially in the early stage. However, this advantage diminishes in validation due to the discrepancy in the distributions. The results will be reported in the revised manuscript.
>
> It is worth a remention that Karanikolos & Refanidis (2019) demonstrated that positional encoding is effective for the text summarization task. It might be the case that positional encoding is only useful when many of the observed tokens become irrelevant (just as in our synthetic benchmark) such as text summarization. Thus, empirical benefits would depend on specific tasks, which will be clarified as limitations in the Discussion.
>
> Finally, we would wish to (re-)emphasize that the contribution of the present study is complementary to investigations of real-world tasks; the destabilization of RNN gradients by infrequent tokens and stabilization by position encoding will never have been identified without the synthetic tasks.
>
> ## Response to the Weaknesses pointed out
>
> 1. Please refer to our response to the requested changes above. While the explorations of the present study is never comprehensive due to the limitations of time and computational resources, the two additional benchmarks would improve this aspect.
> 2. Please refer to our response to the requested changes above.
> 3. The specific focus on the sinusoidal encoding is indeed a limitation of this study, and will be clarified in the Discussion as an open question in future studies. Specifically, explorations of the present study are limited to the specific frequencies and phases designed for Transformers, and it looks more promising to first explore other parameters of the sinusoidal encodings than to study radically different implementations. Since such broad investigations consume more computational resources, they are assigned a low priority than the additional experiments requested by the reviewers.

---

> > ### Comment · Reviewer_7qMR · 2024-10-07
> >
> > I thank the authors for their detailed rebuttal. While I understand the goals of the study, addition of real-world task (wikitext-103 should be fine). Helps calibrate a lot on what is the utility and translation of the idea.
> >
> > Overall, I think the paper is good and of general interest in the community and support acceptance (while not absolutely championing it).

---

### Review · Reviewer_7TzK · 2024-09-25

**Summary Of Contributions:**

This paper studies the impact of positional encoding with Recurrent Neural Networks, to assess whether it would only constitute redundant information (given that the ordering of the input data is explicitly modeled by RNNs) or, conversely, be beneficial.

The authors do so with synthetic experiments: reverse-ordering, (+ delayed addition), and sorting of integers samples from "vocabularies" of various sizes. The experiments encompass both GRUs and LSTMs designs.

Upon examination of the findings, the study highlights that, even though positional encoding could intuitively be seen as irrelevant for RNNs, it does indeed help them (in the scope of the tasks examined). The study further highlights a possible explanation by identifying a strong stabilization of the gradients of non-frequent tokens when positional encoding is included in the learning recipe.

**Audience:**

Yes

**Broader Impact Concerns:**

I do not see unaddressed ethical / impact concerns about this paper.

**Claims And Evidence:**

Yes

**Requested Changes:**

I will follow up on the "weaknesses" bullet points above. Specifically:
1. It'd be nice to pair the results of the paper with metrics accounting for the order when measuring the similarity between two sequences. The RBO [1] is one such example, but I would not necessarily champion this metric.
2. Could the authors please highlight the difference, in terms of the raw number of learnable parameters, between RNNs with and without positional encoding? If this difference appears large, maybe scaling the total parametrization of "vanilla" RNNs to match those with positional encoding would shed some light on the effect that positional encoding has, especially for tasks where the order shall be deemed irrelevant.

(PS: it may as well be that some underlying reason renders the second point unfeasible and I am not spotting it at the moment. I strongly encourage authors to point this out, if that's the case.)


References:
[1] William Webber, Alistair Moffat, and Justin Zobel. A similarity measure for indefinite rankings. ACM
Transactions on Information Systems.

**Strengths And Weaknesses:**

I genuinely like the setup of the paper. Please read the strengths in the bullet points below.

1. The overall "topic" of the paper is interesting, and deviates from the common ground of using positional encoding only with transformer-based models;
2. The paper is well written;
3. The background (i.e., the functionality of positional encoding and the relevant works) is properly introduced;
4. The experiments are well-designed, and the emerging observations are non-trivial.

My concerns on this paper mostly relate to the following factors:
1. While I guess there would not be much difference, probably evaluating with metrics privileging the ordering would lead to a clearer picture. I see that, at the current stage, the paper employs "token-wise accuracy".
2. I wonder whether some results may be a by-product of additional network parametrization in also having positional encoding, since, as per my understanding, the size of the RNNs of this study is relatively small. In particular, this doubt arises when looking at Appendix A.2, where positional encoding also appears to be a big help to the sorting task, where ordering of the input sequence should be completely irrelevant.

---

> ### Author Response · Authors · 2024-09-30
> **Authors' Response to Reviewer 7TzK**
>
> Thank you for your insightful feedback. Please find our responses below.
>
> ## Response to the Requested Changes
>
> ### 1. Alternative Metric
> You are right that the accuracy scores reported here were calculated token-wise (similar to the cross-entropy loss), and thus, a single-step delay at some point in the prediction resulted in mismatches thereafter.
>
> Following your suggestions, the Damerau-Levenshtein distance (edit distance defined by insert, delete, substitute, and SWAP operations) was evaluated between the targets and predictions for the reverse-ordering and the sorting tasks. There was no qualitative difference between the two metrics in consequence, but this investigation is definitely important. We really appreviate your suggestion.
>
> ### 2. Equally Sized Models
>
> First of all, let us clarify that the additional learnable parameters of position-encoded RNNs are limited to the input-to-hidden weights (for each gate in GRU and LSTM), doubling their input channels for accommodating positional encoding (the hidden-to-hidden weights were of the same size between vanilla and position-encoded RNNs). Thus, these extra parameters are blind to the input tokens in the forward pass.
>
> Nevertheless, additional experiments are being performed on the reverse-ordering and sorting tasks to balance the number of tunable parameters; specifically, a different implementation of LSTM w/o positional encoding is being investigated by concatenating two copies of the input embeddings. Accordingly, the input-to-hidden weights are doubled (same as in position-encoded LSTM) without changing the dimensionality of the input embeddings themselves.
>
> At the time of this post, three trials of the reverse-ordering task and only one trial of sorting task have been finished (with the vocabulary size set as 16,384). So far, the extra parameters seem not to improve the model performance. Please stay tuned for the ultimate results.
>
> ## Response to the Weaknesses Pointed Out
>
> 1. Please refer to our response to the Requested Changes.
> 2. Please refer to our response to the Requested Changes. We agree that the improvement in the sorting task is unintuitive, but it also supports our argument that positional encoding is more than timestamps. Although we cannot directly test the gradient stability in the sorting task (because inputs interact with one another and we cannot easily build ideally-identical mappings as in the reverse-ordering task), the efficacy of positional encoding for RNNs is probably stabilization of their gradients, rather than timestamping. In this view, it is not impossible that positional encoding improves the order-insensitive memorization. In any case, it is a major limitation of the present study that the mechanism of the gradient stabilization is left as an open question. We will clarify it in the Discussion in the revised manuscript.

---

> > ### Comment · Reviewer_7TzK · 2024-10-07
> > **Thank you!**
> >
> > Dear Authors,
> >
> > I have read over your responses to all other reviewers and me, as well as the updated manuscript. To me, the proposed changes seem sufficient to recommend acceptance for this paper. I have no other major concern.
> >
> > Bests,
> > Reviewer 7TzK

---

### Review · Reviewer_93n9 · 2024-09-26

**Summary Of Contributions:**

This work proposes improving the performance of RNNs to handle sequential/temporal datasets better by adding positional embedding modules to the inputs embeddings. While RNNs inherently have a temporal structure due to their design, the authors use a simple synthetic benchmark to argue that in the cases of large vocabulary and low frequency, gradient instability is an issue. To mitigate this, they propose to use sinusoidal embeddings that are common in transformer. The improvements in accuracy are shown for some synthetic benchmarks such as reversing an array and also gradient stability is shown to have improvement. Some small-scale modifications are also performed, such as modifying the task or changing the sequence length. The target audience are people interested in improving RNNs or using them for task that rely heavily on temporal structure.

**Audience:**

Yes

**Broader Impact Concerns:**

There are no broader impact concerns.

**Claims And Evidence:**

No

**Requested Changes:**

Critical for recommending acceptance: While the work is potentially interesting to some, I would like to see a more convincing reason (either using theoretical ideas or extensive experiments) that using positional embeddings will largely benefit RNNs across a wide variety of tasks.

Not critical: Typo in section 5: carring -> carrying

**Strengths And Weaknesses:**

Strengths:

- This work proposes the idea of using positional encodings in RNNs to improve their capabilities in settings of large vocabulary and low-frequency alphabets. While similar ideas have been studied before, this specific motivation seems new.

Weaknesses:

- The main glaring weakness is the lack of either strong theoretical insights or convincing real-life experimental evidence for the proposed work. The work only studies simple synthetic benchmark tasks such as reversing and sorting and it's not clear how this proposed technique will fare in more real-life applicatios of RNNs such as in NLP.

- The synthetic tasks also essentially involve positionality concepts heavily, such as reversing an array. Therefore, it may possibly be the case that positional embeddings specifically help such narrow tasks whereas be not important in potentially other tasks such as summarization, semantics, classification, etc.

---

> ### Author Response · Authors · 2024-09-30
> **Authors' Response to Reviewer 93n9**
>
> Thank you for your constructive feedback. Please find our responses below.
>
> ## Response to the Requested Changes
>
> We would first wish to remind you that Karanikolos & Refanidis (2019) demonstrated the effectiveness of positional encoding for the text summarization task. And the present study is mainly intended to complement such investigations of real-world tasks; the destabilization of RNN gradients by infrequent tokens and stabilization by position encoding will never have been identified without the control of the vocabulary size and token frequencies in the synthetic tasks. Thus, although we do not have a rigorous, mathematical explanation about the mechanism of the destabilization & stabilization, we think that our findings through the synthetic benchmarks are non-trivial and useful for future studies.
>
> In the mean time, we do agree that explorations of emprical benchmarks are important especially because position-encoded RNNs have been overlooked in the community for four years. We are thus testing language modeling on wikitext-103 (w/ custom tokenization by SentencePiece, due to the limitations of computational resources). The task was selected simply because it is the most canonical in this era of LLM. At the time of this post, four trials (with different random seeds) are done for vanilla and position-encoded LSTMs, and so far, position-encoded LSTMs seem to learn a bit faster (in terms of the drop in training perplexities) especially in the early stage. However, this advantage diminishes in validation due to the discrepancy in the distributions. The results will be reported in the revised manuscript.
>
> ## Response to the Weaknesses pointed out
>
> 1. Please refer to our response to the the Requested Changes.
> 2. As Reviewer 7TzK pointed out, the improvement in the sorting task, which ignores the input order, is unexpected under the canonical view of positional encoding as input timestamps. We will also report an additional synthetic benchmark that is less order-sensitive and more content-sensitive (see our response to Reviewer 7qMR). And, again, we wish to re-shed a light on Karanikolos & Refanidis's (2019) study, which reported the effectiveness of positional encoding in text summarization. We argue that positional encoding contributes to the robustness of RNNs against the gradient instability caused by infrequent tokens, presumably affecting the latent dynamics of the network activity. In this way, positional encoding can be effective in seemingly order-insensitive tasks as well, although the amount of benefits is definitely task-dependent.

---

> > ### Comment · Reviewer_93n9 · 2024-10-13
> > **Thank you for your response.**
> >
> > I thank the authors for their response and updating the manuscript with an experiment with S4 and an additional benchmark. I have also gone over the other reviews and the responses. While I agree that the ideas presented are potentially interesting and worth pursuing further, I still feel the present contributions to be lacking. If real-world applications of positional encodings in RNNs are already presented in prior work, that further lessens the contributions of this work. Moreover, analyzing such techniques on synthetic tasks alone can offer good intuition but will require either novel theoretical insights or extensive experimentation (more than App E) to validate the ideas. For these reasons, I still think more contributions are needed for me to recommend acceptance of this work.
> >
> > Typo in table 1: "obatined" -> "obtained"

---

> > > ### Author Response · Authors · 2024-10-13
> > > **A clarification question regarding the deficiency of contributions**
> > >
> > > Let us first express our gratitude to you for pointing out our typo. We will correct it in the next version of the manuscript.
> > >
> > > Concerning the perceived deficiency of contributions, we wish to ask you a clarification question whether you think that our submission fails to meet the [Acceptance Criteria of TMLR](https://jmlr.org/tmlr/acceptance-criteria.html) (i.e., our claim is NOT supported by accurate, convincing or clear evidence, or nobody is interested in our findings), or our study is deemed insufficiently significant to be certificated as a “Featured” or “Outstanding” paper.
> > >
> > > While our study does not provide rigorous mathematical proofs, we maintain that our experiments "supported" the destabilization of RNN gradients due to infrequent tokens and their stabilization through positional encoding. Moreover, we posit that synthetic benchmarks are more suitable for our objectives, as a collection of real-world benchmarks would merely tell us in which task positional encoding is useful, without facilitating a discussion on token frequency or gradient stability. And we also believe that our findings will be of interest to at least someone in TMLR's audience, despite the current limitations in empirical applications.

---

> > > > ### Comment · Reviewer_93n9 · 2024-10-16
> > > > **Response to rebuttal**
> > > >
> > > > The research direction is definitely of interest to people in the TMLR audience, however my concern is with the amount of convincing/clear evidence shown to support the thesis presented in this work. In the absence of any other contributions, experiments on a couple of synthetic benchmarks alone is somewhat weak to argue that positional encoding convincingly mitigates effects of infrequent tokens in a wide variety of applications. Therefore, the experimental section has to be more comprehensive if that's the primary thrust of this work.

---

> > > > > ### Author Response · Authors · 2024-10-17
> > > > > **Suggestions for additional experiments**
> > > > >
> > > > > Thank you so much for your response. We now understand your perspective.
> > > > >
> > > > > To enhance the present study, we would greatly appreciate your assistance in designing additional experiments. Typical benchmarks utilized in previous studies on new RNN architectures include the copying memory task and sequential MNIST [e.g., 1-5]. In the copying memory task, a model receives a sequence of random integers and must recover it after a designated "stay" period. This task formed the basis of our synthetic benchmarks; we customized it to explore the effects of vocabulary size and token frequency (together with modifications of the output sequence order and additions of algebraic/syntactic operations). By contrast, the sequential MNIST does not involve manipulatable parameters corresponding to "vocabulary size" or "token frequency"; the input sequence consists of 8-bit integers (typically treated as floats), and the output target is a digit.
> > > > >
> > > > > We are seeking tasks that allow for (1) the modification of vocabulary size or token frequency and (2) qualitatively complement the current inventory of experiments (hopefully providing insights for future mathematical analyses). Thus, it would be immensely helpful if you could suggest specific experiments worthy of investigation to address the gap in the current draft.
> > > > >
> > > > > [1] Arjovsky et al. 2016. "Unitary Evolution Recurrent Neural Networks", ICML.
> > > > > [2] Chang et al. 2017. "Dilated Recurrent Neural Networks", NeurIPS.
> > > > > [3] Jing et al. 2017. "Tunable Efficient Unitary Neural Networks (EUNN) and their application to RNNs", ICML.
> > > > > [4] Voelker et al. 2019. "Legendre Memory Units: Continuous-Time Representation in Recurrent Neural Networks", NeurIPS.
> > > > > [5] Gu et al. 2020. "HiPPO: Recurrent Memory with Optimal Polynomial Projections", NeurIPS.

---

### Review · Reviewer_wCde · 2024-09-26

**Summary Of Contributions:**

This paper studies the impact of postional embedding in RNN models and points out that positional embedding can stabilize the graident of RNNs to handle larger vocabularies. The authors design the reverse-ordering task and test GRU and LSTM to analyze the gradient stability to understand the mechanism of improvement. In the experiments, positional encoding was found to improve RNN performance when handling rare tokens surrounded by more frequent tokens, especially in long sequences.

**Audience:**

Yes

**Claims And Evidence:**

Yes

**Requested Changes:**

As noted in the weaknesses, the authors are encouraged to expand the experiments and provide more in-depth discussions. Specifically, it would be beneficial to explore how the findings generalize to larger-scale, real-world datasets and to compare the role of positional encoding in RNNs with its use in state-of-the-art models like Mamba [1]. A deeper examination of the relationship between positional encoding and the mechanisms in models like Mamba would further strengthen the paper.

Overall, the paper presents a novel and valuable contribution. However, more comprehensive experiments and a more thorough discussion would enhance its impact.

[1] Gu, Albert, and Tri Dao. "Mamba: Linear-time sequence modeling with selective state spaces." arXiv preprint arXiv:2312.00752 (2023).

**Strengths And Weaknesses:**

Strengths:
1. The authors present an unexpected and significant discovery about the benefits of positional encoding (PE) for RNNs, which challenges the existing assumptions about RNNs' inherent ability to handle temporal information without external mechanisms like PE.
2. The design of task and model architecture is simple and clear, which helps to understand the underlying mechanisms in depth.

Weaknesses:
1. Although the paper mentions other tasks in the appendix, all experiments are conducted on synthetic data. While these toy examples are useful for exploring potential mechanisms, it raises the question of whether the findings can generalize to larger-scale, real-world data.
2. The related works section could be expanded to include a discussion on state-space models and models like Mamba. It would be valuable to explore the differences and connections between the role of PE in these models versus RNNs.
3. In the appendix, the authors present results from multiple implementations of positional embeddings, showing that random embeddings outperform learned ones. This aspect needs further discussion. Additionally, it would be beneficial to include a comparison of absolute, relative, and other forms of positional embeddings to enrich the analysis.

---

> ### Author Response · Authors · 2024-09-30
> **Authors' Response to Reviewer wCde**
>
> Thank you for your constructive feedback. Please find our responses below.
>
> ## Response to the Requested Changes
>
> ### State Space Model
>
> Following your suggestion, we are now testing the S4(D) (Gu et al., 2022, ICLR/NeurIPS) on the reverse-ordering task (with the vocabulary of size 16,384; we decided to start with a simpler model than Mamba due to its ease of implementation). At the time of this post, we have finished one trial for the vanilla and position-encoded models. While it might be too early to make any conlusion right now, **positional encoding seems to be effective for S4 as well**. That is, the vanilla model only achieved the test accuracy of 56.76% while the position-encoded model did 72.38%. Please stay tuned for the ultimate results.
>
> The revised manuscript will be extended to include these results on S4, covering as many experiments as we can in the time limit. If positional encoding indeed turns out to be effective for the state space model, it will become the most important update in the revisions, suggesting the relevance of our findings to modern models. We really appreciate your insightful comments directing our eyes to them.
>
>
> ### Real-World Benchmark
>
> We would first wish to remind you that Karanikolos & Refanidis (2019) demonstrated the effectiveness of positional encoding for the text summarization task. And the present study is mainly intended to complement such investigations of real-world tasks; the destabilization of RNN gradients by infrequent tokens and stabilization by position encoding will never have been identified without the synthetic tasks.
>
> Nonetheless, we do agree that explorations of emprical benchmarks are important especially because position-encoded RNNs have been overlooked in the community for four years. We are thus testing language modeling on wikitext-103 (w/ custom tokenization by SentencePiece, due to the limitations of computational resources). The task was selected simply because it is the most canonical in this era of LLM. At the time of this post, four trials (with different random seeds) are done for vanilla and position-encoded LSTMs, and so far, position-encoded LSTMs seem to learn a bit faster (in terms of the drop in training perplexities) especially in the early stage. However, this advantage diminishes in validation due to the discrepancy in the distributions. The results will be reported in the revised manuscript.
>
> It might be the case that positional encoding is only useful when many of the observed tokens become irrelevant (just as in our synthetic benchmark) such as text summarization. Thus, empirical benefits would depend on specific tasks, which will be clarified as limitations in the Discussion.
>
> ## Response to the Weaknesses Pointed Out
> 1. Please refer to our response to the Requested Changes.
> 2. As explained in our response to the Requested Changes, we are testing S4(D) and it looks that positional encoding also helps the state space model handle a large vocabulary. These results will be included in the revised manuscript, and we will of course extend the Related Studies section as suggested.
> 3. The specific focus on the sinusoidal encoding is indeed a limitation of this study, and will be clarified in the Discussion as an open question in future studies. Specifically, explorations of the present study are limited to the specific frequencies and phases designed for Transformers, and it looks more promising to first explore other parameters of the sinusoidal encodings than to study radically different implementations. Regarding the relative encoding, the distance-representing vectors associated with past tokens must be updated for every increment of time steps. This is not easily compatible with RNNs, which do not re-evaluate the past tokens for each time step like Transformers.

---

> > ### Author Response · Authors · 2024-10-07
> > **Updates regarding the experiment on a state-space model**
> >
> > As anticipated in the previous post, positional encoding indeed improved the accuracy of a state-space model (S4D) on the reverse-ordering task involving a large vocabulary. S4D also exhibited the same pattern of performance degradation as the RNNs; its accuracy declined when Rare disturbant tokens were present in the input sequence.
> >
> > HOWEVER, unlike the RNNs, the gradients of the vanilla S4D remained stable in the presence of these Rare disturbants, despite the observed decline in accuracy. Thus, the analysis of gradients does not offer a unified explanation for the (un)successful learning of RNNs and state-space models. The revised manuscript acknowledges this limitation and notes it as an open question for future research.
> >
> > We also apologize that the revised manuscript still lacks a paragraph for introducing neural state-space models. (We priortized reporting the additional experiments before completing the entire revisions). This will be added to the Related Study section as soon as possible.

---

> > > ### Author Response · Authors · 2024-10-10
> > > **Extension of the Related Studies section**
> > >
> > > The Related Studies section (2.1) was finally extended to cover state-spaced models.
> > > We apologize for this last-minute revision.

---

### Author Response · Authors · 2024-10-01
**Erratum**

This post reports an errortum in the first draft of the manuscript.

In Section 4.2 (and 4.3 as well), where we studied the frequency effect by splitting the vocabulary into Frequent vs. Rare tokens, the reported probability of token sampling was incorrect.

> ... the probability of each Frequent token was ~~3/4~~ **7/8** × 2/$K$ (where $K$ denotes the total vocabulary size and was set to 64 and 1024 for GRU and LSTM, respectively) whilist the probability of each Rare token was ~~1/4~~ **1/8** × 2/$K$.

That is, the Frequent and Rare tokens were more and less frequent respectively than originally reported.
(The parameter value reported in the supplementary material was correct.)

We apologize for this errortum. It will be fixed in the revised manuscript. This revision does not make a qualitative change in the overall argument of the section.

---

### Decision · Action_Editor_feZp · 2024-11-08

**Recommendation:** Accept with minor revision

**Comment:**

The main concern of reviewers on acceptance of the paper was the experimental assessment performed only on synthetic data. The author, however, was able to improve this aspect adding more empirical evidence  in the revised version, also providing further discussion on the characterisation of the proposed approach. Overall, 3 out of 4 reviewers judged the current version of the paper sound and interesting for the TMLR audience.
The content of the paper does not fit any of the certifications.
The current version of the paper needs to fix some formatting issues, such as text overflow into right-hand margin.

**Audience:**

Given the relevance of RNN architectures in many ML applications, the covered topic is for sure of interest for TMLR's audience.

**Claims And Evidence:**

The revised version of the paper after the review seems to satisfy 3 out of 4 reviewers. Although the experimental assessment is not very strong (as noted by one reviewer), it seems to me that it is sufficient to support the author's claim that positional encoding may be beneficial for RNN. Of course, there will be specific tasks (i.e., positional invariant/equivariant tasks) where positional encoding may not help learning in RNN, however if presence/absence of the positional encoding is considered as an additional hyper-parameter for model selection, then I can see the potential advantage of considering it.